# Plant Growth Promotion and Biocontrol of *Pythium ultimum* by Saline Tolerant *Trichoderma* Isolates under Salinity Stress

**DOI:** 10.3390/ijerph16112053

**Published:** 2019-06-10

**Authors:** Brenda Sánchez-Montesinos, Fernando Diánez, Alejandro Moreno-Gavira, Francisco J. Gea, Mila Santos

**Affiliations:** 1Departamento de Agronomía, Escuela Superior de Ingeniería, Universidad de Almería, 04120 Almería, Spain; brensam@hotmail.com (B.S.-M.); fdianez@ual.es (F.D.); alejanmoga@gmail.com (A.M.-G.); 2Centro de Investigación, Experimentación y Servicios del Champiñón (CIES), Quintanar del Rey, 16220 Cuenca, Spain; fjgea.cies@dipucuenca.es

**Keywords:** *Pythium ultimum*, salt tolerance, *Trichoderma*, biological control, stress abiotic and biotic

## Abstract

This present study evaluates three isolates of *Trichoderma* as plant growth promoting or biological control agents: *Trichoderma aggressivum* f. sp. *europaeum*, *Trichoderma saturnisporum*, and the marine isolate obtained from *Posidonia oceanica*, *Trichoderma longibrachiatum*. The purpose is to contribute to an overall reduction in pesticide residues in the fruit and the environment and to a decrease in chemical fertilizers, the excess of which aggravates one of the most serious abiotic stresses, salinity. The tolerance of the different isolates to increasing concentrations of sodium chloride was evaluated in vitro, as well as their antagonistic capacity against *Pythium ultimum*. The plant growth promoting capacity and effects of *Trichoderma* strains on the severity of *P. ultimum* on melon seedlings under saline conditions were also analysed. The results reveal that the three isolates of *Trichoderma*, regardless of their origin, alleviate the stress produced by salinity, resulting in larger plants with an air-dry weight percentage above 80% in saline stress conditions for *T. longibrachiatum*, or an increase in root-dry weight close to 50% when *T. aggressivum* f. sp. *europaeum* was applied. Likewise, the three isolates showed antagonistic activity against *P. ultimum*, reducing the incidence of the disease, with the highest response found for *T. longibrachiatum*. Biological control of *P. ultimum* by *T. aggressivum* f. sp. *europaeum* and *T. saturnisporum* is reported for the first time, reducing disease severity by 62.96% and 51.85%, respectively. This is the first description of *T. aggressivum* f. sp. *europaeum* as a biological control agent and growth promoter. The application of these isolates can be of enormous benefit to horticultural crops, in both seedbeds and greenhouses.

## 1. Introduction

The growing concern of consumers for food safety and for the social and environmental sustainability of cultivation systems has a particular impact on the fruit and vegetable production sectors. These sectors are facing increasingly stringent restrictions that large distribution chains establish in their purchasing specifications regarding the presence of active ingredients in multi-waste pesticide analyses. These are much more restrictive than those established by community legislation itself, both in terms of the amount (less than the maximum residue limit, MRL) and in terms of the number of active substances detected that they permit [1]. This circumstance forces us to look for new tools for crop protection that are not based on chemical control. Currently, in important areas of intensive vegetable production such as Almería (Spain), there has been a notable advance in the biological control of pests, but not in that of diseases, the latter still being quite dependent on the use of agrochemicals. The use of microorganisms as biological control agents (BCAs) of diseases is one of the keys to production with less phytosanitary residues and greater food safety. In addition, many of the BCAs marketed act as promoters of plant growth. This circumstance permits action on another of the important problems of intensive production systems, environmental pollution due to the excessive use of fertilizers. The search for new tools to reduce the use of pesticides and chemical fertilizers is a goal that must be achieved in the medium term.

The productivity of agricultural lands in arid and semi-arid environments is affected by the accumulation of salts and the loss of soil organic matter [2]. The salinity of the soil, as well as the use of water with a high salt content for irrigation, leads to a reduction in plant growth and crop yields [3,4]. Likewise, the presence of salts affects the assimilation of nutrients by plants and the microbial activity of the soil. In addition, it increases the severity of the effects of phytopathogens and influences biological control agents and the interaction among them. There are numerous references to the buffering role of microorganisms in the plant rhizosphere [4,5,6]. Generally, this effect is due to molecular, biochemical and physiological changes in the plant elicited by microorganisms [7]. *Paecilomyces formosus* mitigates the negative impact of salt stress on cucumber plants by producing gibberellins and indoleacetic acid [8]. Likewise, the association with endophytic fungi such as *Penicillium* sp. and *Phoma glomerata* alters jasmonic acid levels, increases salicylic acid values and reduces abscisic acid synthesis, reducing the detrimental effects of salinity [9]. Coinoculation of *Aspergillus niger* and *Trichoderma harzianum* alleviates the deleterious effects of salt stress on wheat seedlings through the solubilisation of P and joint production of indoleacetic acid [10]. Yasmeen and Siddiqui [11] detected that the presence of *Trichoderma* in a saline environment increased the activity of antioxidant enzymes. Studies carried out with *T. harzianum* confirmed its high soil colonisation and yield in the production of tomatoes grown under plastic with high-salinity irrigation [12]. Nevertheless, there are microorganisms that are not adapted to conditions of high salinity. Such is the case for some isolates of *Trichoderma*, whose biofungicide and plant promoter role may be compromised by their low osmotolerance [13,14]. For this reason, numerous studies have been carried out in the search for halotolerant biological control agents. Gal-Hemed et al. [15] isolated *T. atroviride* and *T. asperelloides* from the Mediterranean sponge *Psammocinia* sp. capable of reducing *Rhizoctonia solani* damping-off disease in beans and also inducing defence responses in cucumber seedlings against *Pseudomonas syringae* pv. *lachrymans*. Likewise, *Trichoderma* isolates of marine sediments, invertebrates and algae have been obtained [16,17]. On the other hand, breeding of *Trichoderma* has been conducted, with the same objective of increasing the benefits of biological control, as well as resisting adverse conditions [13].

Therefore, the goals of the present study were: (a) to evaluate the tolerance of different concentrations of NaCl on three isolates of *Trichoderma* obtained from suppressive soils, mushroom culture substrates, and *Posidonia oceanica*; (b) to evaluate the in vitro antagonistic capacity of these isolates against *Pythium ultimum*; (c) to study the capacity to promote the growth of melon seedlings of *Trichoderma* strains under increasing salinity concentrations; and finally, (d) to evaluate the biological control of the disease caused by *P. ultimum* in melon seedlings, exerted by *Trichoderma* strains under various levels of salt stress.

## 2. Materials and Methods

### 2.1. Fungal Isolates

The isolates selected in this study have been obtained from different environments. *Trichoderma saturnisporum*—(TS), obtained from suppressive soils, was selected for its known antagonistic activity and plant growth promotion in pepper and melon [18,19]. *Trichoderma aggressivum* f. sp. *europaeum* Tae52481 (TA) was isolated from samples of substrate used for *Agaricus bisporus* cultivation from mushroom farms located in Castilla-La Mancha (Spain). *Trichoderma longibrachiatum* (TL) was isolated from the roots of the endemic seagrass, *Posidonia oceanica*, sampled from the Mediterranean Sea in Almeria, Spain, following the procedure of Vohník et al. [20]. Both were isolated by serial dilution technique on potato dextrose agar (PDA) medium and incubated at 25 °C for 5–7 days.

Colony morphology of the pure cultured isolates TL and TA on potato dextrose agar (PDA, Difco) and conidiophore morphology examined by light microscopy and cryo-fracturing electron scanning microscopy (cryo-sem) [18] were consistent with the genus *Trichoderma* (Figure 1). Molecular identification of the selected fungi was conducted following the procedure described by Diánez et al. [18]. The sequence was analysed using a BLAST search in the GenBank database of the National Centre for Biotechnology Information (NCBI, http://blast.ncbi.nlm.nih.gov/Blast.cgi) and aligned to the nearest neighbours. The sequences have not been deposited in the GenBank database as they are subject to patent.

Melon seedlings showing lesions caused by *Pythium ultimum* were selected to obtain the inoculum, which was used as the test pathogen in antagonistic in vitro and in vivo assays.

Each isolate was grown on PDA for 5 or 15 days at 25–27 ± 2 °C under dark conditions. Spore suspensions of *Trichoderma* isolates were prepared by flooding plates of 15-day-old cultures with sterile distilled water, scraping with a sterile glass rod and filtering, and adjusted to a concentration of 1·10^8^ spores/mL with a Neubauer haemocytometer.

### 2.2. Growth Conditions of the Fungal Strains in PDA with Differing Concentrations of NaCl

*Trichoderma* and phytopathogen strains were cultured at 25 °C for 7 days in PDA. Mycelium discs (5 mm diameter) were obtained from the edges of the PDA plates and inoculated into new PDA, amended with various concentrations of NaCl (0, 1, 2, 5, 10, 15 or 20 g·L^−1^). The plates were incubated for 7 days at 25 and 35 °C. The colony diameters were recorded every day during culturing. The experiment was completely randomised with five replicates. Salt tolerance capacity of *Trichoderma* isolates was measured as the percentage reduction in linear growth calculated by (C − N)·100/C, where N is the maximum radius of the isolates grown on NaCl-supplemented medium and C is the radius of the isolates grown on NaCl-free medium.

### 2.3. Dual Culture Antagonism Assays

*Trichoderma* isolates were screened for their antagonism against *Pythium ultimum* by the confrontation assay of Santos and Diánez [21]. Petri dishes (9 cm diameter) containing 15 mL of PDA were prepared, amended with different concentrations of NaCl (0, 1, 2, 5, 10, 15, or 20 g·L^−1^). Petri dishes were sealed with parafilm and incubated in the dark at 25 °C for 4–7 days, until the growth in the control plates reached the edge of the plates. The plates were then assessed by measuring the distances between pathogen and fungal cultures. Results were transformed into percentages of mycelium growth inhibition. These tests were carried out in quintuplicate.

### 2.4. Evaluation of Growth Promotion Effects of Trichoderma Isolates on Melon Seedlings under Salinity Stress

To determine the promoter effect of the different isolates of *Trichoderma*, melon seeds of the variety Piñonet (Piel de sapo) were disinfected with 2% hypochlorite for 3 min and washed abundantly with tap water to eliminate residues. Subsequently, the seeds were pregerminated in darkness in a humid room at 25 °C and transplanted to 300 mL pots into a commercial peat mix, with one seed per pot. Simultaneously to sowing, 5 mL of water (T0) or 5 mL spore suspension of each isolate of *Trichoderma* (TS, TA, or TL) was placed in each pot at 50 × 10^6^ propagules/plant. The experiment was performed under greenhouse conditions. Each treatment consisted of 25 repetitions. Plants were fertilized daily with a commercial complex nutrient fertilizer. When the first true leaf had fully expanded, four different levels of NaCl concentrations, 0, 0.5, 1, 1.5 or 2 g·L^−1^, were given by manually drenching the media with approximately 50 mL of the solutions, once per day for 30 days, and with 100 mL per day for the following days, as the plants grew. The electrical conductivity (EC) of the solutions was recorded to be 2.1, 3.87, 5.30, 6.6 and 7.16 dS·m^−1^. After 45 days of culture, 10 plants per treatment and control were analysed. Dry and fresh weight of the aerial part and roots were determined.

### 2.5. Effects of Trichoderma Strains on the Severity of Pythium Ultimum in Melon Seedlings under Saline Conditions

To obtain zoospores of *Pythium ultimum*, the procedure described by Marin et al. was followed [22]. The concentration of the zoospore suspension was adjusted to approx. 10^3^ zoospores·mL^−1^, using a haemocytometer. The inoculum was used immediately, and 5 mL of the suspension was applied uniformly over the surface of the peat in each pot using a sterile micropipette. The pathogenicity test of *P. ultimum* on melon seedlings was carried out under greenhouse conditions, in the same manner as described above. The application of *P. ultimum* was performed after complete expansion of the second true leaf. The application of the different salt concentrations was carried out in the same manner as for the growth promotion test described above. Symptom severity was rated periodically, and a final disease severity index was estimated according to the following scale [23]: 0—healthy plant; 1—symptoms beginning; 2—moderate symptoms; 3—severely affected plant; and 4—dead plant. The experiments were conducted using completely randomised block designs.

### 2.6. Statistical Analysis

All data presented are the mean of five replicates of in vitro tests and ten replicates of pot experiments. Data were analysed using analysis of variance, conducted using the Statgraphics Centurion ver. XVI software. Results are expressed as mean value ± standard error of the mean. *p*-values less than 0.05 were considered to be significant.

## 3. Results

### 3.1. Effects of Salinity and Temperature on Colony Growth of Trichoderma Isolates

In Figure 2, the results obtained from mycelial growth of TS, TA, and TL are shown at different salt concentrations and temperatures (25 and 35 °C). As expected, there was no inhibition of TL mycelial growth at the different salt concentrations tested, nor were there differences in growth at 25 and 35 °C. However, for both TS and TA, the presence of salt in the medium conditioned growth. This resulted in a drastic reduction of mycelial growth, from 66.9 (TS) and 76.75% (TA) from 2 and 10 g·L^−1^, respectively. The temperature of 35 °C completely inhibited the mycelial growth of TA. No reduction in growth of *P. ultimum* was observed at 25 °C as salt concentration increased, except at 20 g·L^−1^. No growth at 35 °C (Figure 2) was observed.

### 3.2. Effects of Trichoderma Isolates on the Radial Growth of P. ultimum

In Figure 3, the results obtained from the microbial antagonism for *P. ultimum* are observed. The highest percentages of inhibition of mycelial growth corresponded to the confrontations with TL, which were not affected under conditions of increasing salinity. The decrease in growth of TA and TS antagonist isolates had an impact on the microbial antagonism detected as the concentration of medium salt increased. Despite this, microbial antagonism values were high, considering the characteristic mycelial growth rate of this pathogen at any salt concentration.

### 3.3. Promoter Effects of Trichoderma Isolates on Melon Seedlings and Salinity Treatments

The effect of *Trichoderma* isolate application by irrigation on morphological parameters is shown in Table 1. The application of TS, TL and TA resulted in increases of both the aerial and radical part of the plant. These increases were statistically significant in some cases. Such promotion of plant growth also occurs under conditions of saline stress.

The highest values were detected for TS at 0 g·L^−1^, where there was an increase in the fresh and dry weight of the aerial and radical parts of 17.5, 62.11, 44.02 and 46.51%, respectively. For the different salinity concentrations tested, plant growth promotion was also observed in the dry weight of the aerial part, which decreased as the salt content in the water increased (from 50.13% at 0.5 g·L^−1^ to 17.16% at 2 g·L^−1^). However, there was a very marked decrease in the root, going from a 5.4% increase in the root at 0.5 g·L^−1^ to a 20% decrease at 2 g·L^−1^, relative to that of the control.

The application of TL also promoted melon seedlings growth, increasing the dry weight of the aerial part by 58.17, 87.94, 94.92, 27.10 and 12.12% in NaCl concentrations of 0–2 g·L^−1^, respectively. Likewise, an increase in the dry weight of the root was also observed for all the treatments tested, reaching a maximum of 22% for a concentration at 1.5 g·L^−1^.

In the case of the application of TA, there was an increase in both the aerial part and the very important radical part, relative to that of the control, in all the treatments tested. Thus, increases of 48, 80.54, 93.90, 61.89, and 32.72% were obtained for the dry weight of the aerial part, and increases of 41.80, 16.21, 56.92, 32.35, and 0%, for the dry weight of the root, for NaCl concentrations between 0 and 2 g·L^−1^, respectively.

### 3.4. Effects of Trichoderma Strains on the Severity of Pythium ultimum in Melon Seedlings under Saline Conditions

The application of sodium chloride in irrigation water has not led to a significant increase (*p =* 0.4699) in the symptoms caused by *P. ultimum* in melon seedlings relative to that of the control (0 g·L^−1^). As determined in greenhouse experiments, three strains (TA, TS and TL) significantly reduced the disease index of *P. ultimum* root rot in melon seedlings, compared to that achieved by the control T0, for the different NaCl concentrations tested (Figure 4).

Thus, at 0 g·L^−1^, the severity of the disease was reduced by 74% by the marine isolate TL, followed by TS and TA, with readings of 51.85% and 62.96%, respectively, relative to that of the control (*p* = 0.0086). There were no significant differences in the ability to control the onset of symptoms in the three isolates of *Trichoderma* tested. In general, as salt content in irrigation water increased, there were no significant differences in the control of disease caused by *P. ultimum* for the three isolates of *Trichoderma* assayed, significantly reducing the symptoms with respect to the control (T0). TL performed greater control of the development of the disease, reaching a reduction of 44.11% for 2 g·L^−1^ of NaCl.

## 4. Discussion

The capacity of using fungal isolates for biological control of diseases or promotion of the growth of plants under cultivation conditions may be conditioned by management, soil conditions or fertigation, temperature, salinity and the presence of heavy metals or pesticides, amongst many other factors. The saline conditions, caused in many cases by excessive chemical fertilisation in intensive horticulture, may call into question the effectiveness of these BCAs. In this study, we have analysed the capacity of isolates of *Trichoderma* obtained from different environments to promote the development of melon seedlings and to control root rot caused by *P. ultimum* under saline stress conditions. Isolates from mushroom culture substrates and rhizomes of *Posidonia oceanica* were identified as *T. aggressivum* f. sp. *europaeum* and *T. longibrachiatum*, respectively.

With the exception of TL, the mycelial growth of TA and TS has been influenced by salinity and temperature. The relationship between temperature and the development of *Trichoderma* depends in many cases on the species and the origin of the isolation. Thus, TA showed considerable differences in growth, depending on temperature (25 and 35 °C). Similar results were obtained by Sobieralski et al. [24], whose isolates of *T. aggressivum* f. *europaeum* exhibited very poor growth at a temperature of 35 °C. Both TS and TA drastically reduced mycelial growth in a saline medium. Given the origin of TA, it is logical to presume it has a low tolerance to the presence of salt in the medium. However, this is not so for TS, since it was isolated from sand in the seabed of the coast of Cadiz (Spain), which showed suppressiveness against *Fusarium oxysporum* f. sp. *dianthi* [25]. At 30 g·L^−1^, its growth and sporulation was optimal after its isolation (Diánez F., personal communication), although this growth capacity was reduced after its growth in vitro. De la Cruz et al. [26] consider that there is no significant correlation between marine habitat and salt tolerance of fungal isolates, so it is unclear why marine fungi have different degrees of tolerance to salt, or why they lose that tolerance.

We have shown that TA, TS, and TL can inhibit the development of *P. ultimum* in vitro and are effective in reducing disease severity in melon seedlings, even under saline stress conditions. Given the growth capacity of *P. ultimum* at concentrations of 20 g·L^−1^, it is necessary to apply saline stress tolerant microorganisms that do not lose their antagonistic capabilities against phytopathogens. Migheli et al. [27] showed the capacity of *T. longibrachiatum* CECT2606 to reduce the incidence of damping-off in cucumber, and the role of cellulases in the control of this disease. Recently, Yuan et al. [28] showed the capacity of *T. longibrachiatum* H9 as a growth promoter of cucumber plants and as BCA, reducing the disease index of gray mold caused by *Botrytis cinerea* by induced resistance. TL has been termed probiotic thanks to the multiple benefits to the associated host tomato plants and in disease control [29]. Plant probiotic microorganisms (PPM), also known as bioprotectants, biocontrollers, biofertilizers or biostimulants, are beneficial microorganisms that provide an alternative to the use of pesticides and fertilizers, by reducing environmental and public health problems [30].

*Trichoderma saturnisporum* has been described as a biostimulant in pepper, melon and *Arabidopsis thaliana* [18,19,31] plants, as well as BCA against *Phytophthora parasitica*, *P. capsici* [18] and *Fusarium oxysporum* [31]. New marine TS isolates from *Dictyonella incisa* sponge have recently been described, characterising new secondary metabolites of this species involved in disease control [32].

For the first time, this paper describes the promotion of plant growth and the control of *P. ultimum* by means of *Trichoderma aggressivum* f. sp. *europaeum*. In Europe, this fungus produces very serious decreases in mushroom yield [33,34]. There are no references to this fungus as a phytopathogenic agent. Its high mycelial growth and sporulation, as well as its high power as a pathogen, make TA a promising BCA.

There are numerous references based on the important role played by different species of *Trichoderma* in both plant promotion and biological disease control. The characterisation of the mechanisms involved is well studied and in continuous development [35,36,37]. However, in intensive horticulture under plastic, the benefits of the application of biostimulants or biofungicides based on *Trichoderma* or other microorganisms are in question, due to the perception that farmers have of the low efficacy of these products as disease controllers when compared with the rapid response presented by a chemical fertilizers or fungicide. The current changes in legislation regarding the reduction of active ingredients [38] and the commercialisation of biostimulants and biopesticides, together with the need to increase the sustainability of agriculture in terms of public health and the environment, require the use of PPM as a key element in intensive horticulture.

## 5. Conclusions

The three isolates of *Trichoderma* studied have shown different degrees of tolerance to the presence of NaCl, demonstrating antagonism in vitro against *P. ultimum*. The marine isolate *T. longibrachiatum* did not lose antagonist activity at high salt concentrations (20 g·L^−1^). The present results clearly demonstrate that *Trichoderma aggressivum* f. sp. *europaeum*, *T. saturnisporum* and *T. longibrachiatum* were effective promoters of plant growth and reduced the rate of radicular putrification caused by *P. ultimum* in melon seedlings under saline stress conditions. This paper is the first description of *Trichoderma aggressivum* f. sp. *europaeum* as a plant promoter and BCA.

## 6. Patents

The isolated *Trichoderma aggressivum* f. sp. *europaeum* has been deposited in the CECT and is undergoing patentability studies, with nEPMO reference number P201731151.

## Figures and Tables

**Figure 1 ijerph-16-02053-f001:**
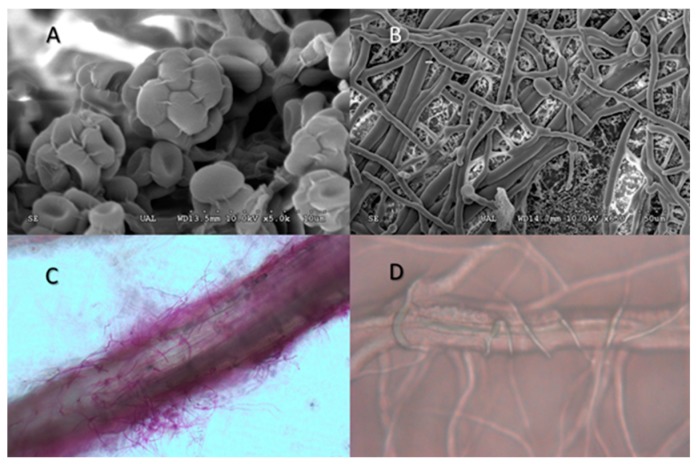
*Trichoderma isolates* examined by electron and light microscopy. (**A**) Conidiophores with conidia of *Trichoderma longibrachiatum*; (**B**) Hyphae of *Trichoderma saturnisporum*; (**C**) *Trichoderma aggressivum* colonisation on melon root. (**D**) *T. aggressivum* mycelia around mycelium of *Pythium*
*ultimum*.

**Figure 2 ijerph-16-02053-f002:**
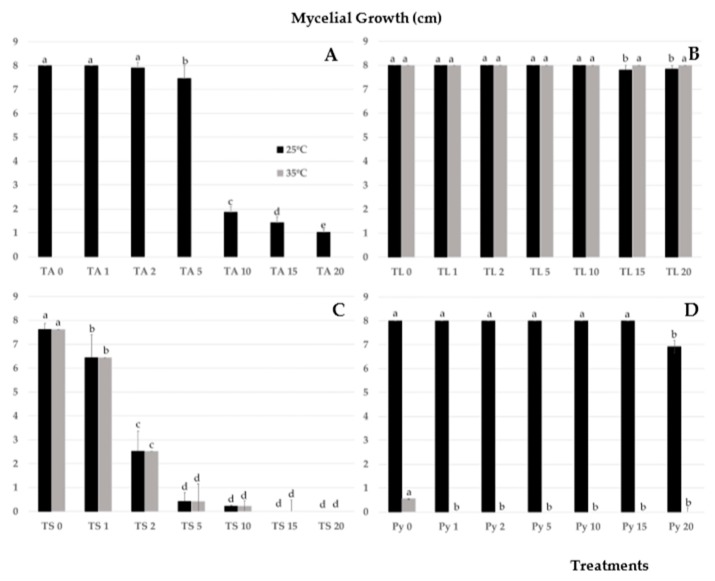
Mycelial growth (cm) of *Trichoderma* isolates as affected by different concentrations of NaCl (0–20 g·L^−1^) and temperatures (25/35 °C). (**A**) *Trichoderma aggressivum* f. sp. *europaeum*. (**B**) *T. longibrachiatum*. (**C**) *T. saturnisporum*. (**D**) *Pythium ultimum*. Mean standard deviation is expressed in error bar (*n* = 5). For each isolate, columns marked with different letters indicate a significant difference at *p* < 0.05.

**Figure 3 ijerph-16-02053-f003:**
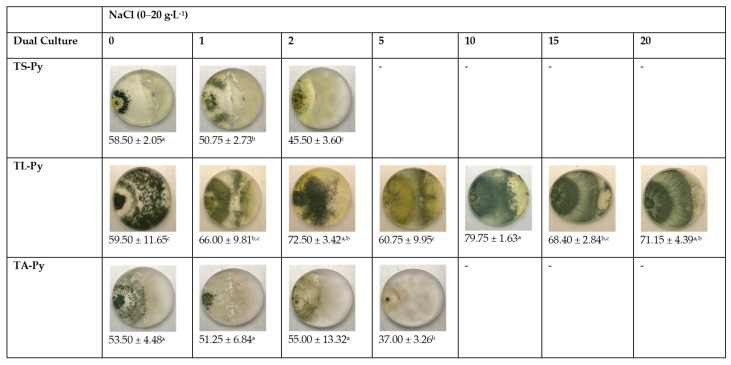
Antagonistic potential of *Trichoderma* isolates against *P. ultimun* (Py) in dual culture at different salinity levels on potato dextrose agar (PDA) medium. % mycelial inhibition was calculated as PIRG = (R1 − R2) ÷ R1 × 100, where: PIRG: percentage inhibition of radial mycelia growth of the pathogen, R1: radial growth of pathogen in control plates, R2: radial growth of pathogen in dual culture plates. ^a, b and c^ Means with the same letter are not significantly different (LSD) according to ANOVA test (*p* < 0.05).

**Figure 4 ijerph-16-02053-f004:**
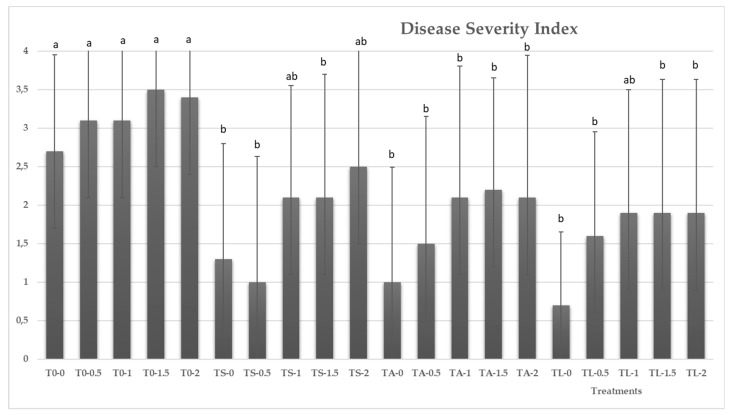
Disease incidence of *P. ultimum* in melon plants was rated 30 days after inoculation based on a 0–4 scale: where 0 = no visible disease symptoms and 4 = plant dead. Mean standard deviation is expressed in error bar (*n* = 10). ^a, b, c^ Means with the same letter are not significantly different (LSD) according to ANOVA test (*p* < 0.05).

**Table 1 ijerph-16-02053-t001:** Morphological parameters of melon plants treated with different doses of NaCl and *Trichoderma* isolates.

Treatments Isolate/NaCl (g·L^−1^)	Aereal Fresh Weight (g)	Root Fresh Weight (g)	Aereal Dry Weight (g)	Root Dry Weight (g)
**T0-0**	6.74 ± 1.93 ^c^	1.61 ± 0.70 ^b,c^	0.48 ± 0.15 ^b^	0.09 ± 0.06 ^b^
**TS-0**	7.92 ± 1.79 ^a^	2.31 ± 0.50 ^a^	0.78 ± 0.16 ^a^	0.13 ± 0.02 ^a^
**TA-0**	7.01 ± 0.98 ^b,c^	1.59 ± 0.85 ^b,c^	0.71 ± 0.12 ^a^	0.12 ± 0.03 ^a^
**TL-0**	7.27 ± 1.71 ^b^	1.74 ± 0.49 ^b^	0.76 ± 0.07 ^a^	0.10 ± 0.04 ^a,b^
**T0-0.5**	6.94 ± 1.80 ^b^	1.42 ± 0.42 ^b^	0.37 ± 0.09 ^c^	0.07 ± 0.03 ^b^
**TS-0.5**	7.19 ± 1.77 ^a,b^	1.68 ± 0.54 ^a^	0.55 ± 0.11 ^b^	0.08 ± 0.02 ^a,b^
**TA-0.5**	7.49 ± 1.34 ^a^	1.70 ± 0.61 ^a^	0.66 ± 0.16 ^a,b^	0.09 ± 0.02 ^a^
**TL-0.5**	7.05 ± 1.22 ^b^	1.39 ± 0.48 ^a,b^	0.69 ± 0.23 ^a^	0.08 ± 0.03 ^a,b^
**T0-1**	5.32 ± 1.45 ^b^	0.93 ± 0.55 ^b^	0.39 ± 0.18 ^b^	0.07 ± 0.03 ^a,b^
**TS-1**	6.01 ± 1.30 ^a^	1.02 ± 0.35 ^b^	0.49 ± 0.11 ^a,b^	0.06 ± 0.02 ^c^
**TA-1**	5.90 ± 1.05 ^a,b^	1.46 ± 0.42 ^a^	0.76 ± 0.12 ^a^	0.10 ± 0.01 ^a^
**TL-1**	6.30 ± 2.43 ^a^	1.35 ± 0.46 ^a^	0.77 ± 0.16 ^a^	0.07 ± 0.03 ^a,b^
**T0-1.5**	5.26 ± 1.40 ^c^	0.67 ± 0.35 ^c^	0.39 ± 0.09 ^c^	0.07 ± 0.04 ^a,b^
**TS-1.5**	5.57 ± 1.03 ^b^	1.02 ± 0.24 ^b^	0.50 ± 0.15 ^b^	0.05 ± 0.02 ^b^
**TA-1.5**	7.43 ± 1.79 ^a^	1.27 ± 0.25 ^a^	0.63 ± 0.18 ^a^	0.09 ± 0.02 ^a^
**TL-1.5**	5.07 ± 0.87 ^c^	1.04 ± 0.23 ^b^	0.50 ± 0.11 ^b^	0.08 ± 0.02 ^a^
**T0-2**	5.22 ± 1.12 ^c^	0.55 ± 0.28 ^b^	0.44 ± 0.14 ^b^	0.07 ± 0.03 ^a^
**TS-2**	5.82 ± 0.83 ^b^	0.72 ± 0.28 ^a^	0.51 ± 0.13 ^a^	0.05 ± 0.02 ^b^
**TA-2**	6.35 ± 1.00 ^a^	1.08 ± 0.21 ^a^	0.58 ± 0.08 ^a^	0.07 ± 0.03 ^a^
**TL-2**	5.42 ± 1.18 ^b,c^	0.82 ± 0.33 ^a^	0.49 ± 0.12 ^a,b^	0.07 ± 0.03 ^a^

^a, b, c^ Values of a column followed by the same letters have no significant difference at 5% (LSD test).

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
