# Peer review of "Plant Growth Promotion and Biocontrol of *Pythium ultimum* by Saline Tolerant *Trichoderma* Isolates under Salinity Stress"

_ijerph, 2019, doi:10.3390/ijerph16112053_

Round 1
Reviewer 1 Report
Please find the attachment.

Author Response
The figures has been modified and statistical analysis has been included.

Reviewer 2 Report
The manuscript (MS) presents an interesting study where fungal candidates of biological control have been tested under salinity stress. The approach of using pathogen of cultivated fungi against plant pathogen is novel. The applicability of the results in practical fruit and vegetable production is very promising.
The MS starts with a good general description of greenhouse production, its problems and promises of biological and integrated pest management. It is very informative and nicely links the study in practice. However, I consider that strong arguments, such as distribution chains establishing “much more restrictions than those to community legislation itself”, would require references for support. It is noteworthy that in the end of MS, similar statements such as “current changes in legislation” and “need to increase sustainability of agriculture in terms of public health and environment” are again given with no references. I do believe that the described processes are ongoing, but the authors should provide explanation and their source of information.
I like the novel idea of looking for antagonists among pathogens of fungi. For some reason, I found no attempts of explaining the background of the approach in the introduction of the MS. General information of combative fungal interactions surely had been available in the literature.
A problem with the MS is that it does not discuss the modes of interaction between plant and Trichoderma species nor fungal interactions. Figure 1 is very interesting as it shows Trichoderma aggressivum colonization on melon root. I am not aware of the possible mode of colonization, athough Trichoderma appears to promote plant growth. The authors could further clarify the interaction type, if it is known, or shortly review literature of such interactions. Also there is accumulating information of fungal interactions which could be used in explaining the antagonism type of reactions described here.
The presentation of the results could be improved. The font size in figures 2,3 and 5 is too small to allow detailed examination in printed form. The different Trichoderma isolates could be illustrated with different bar color or splitting figures in a, b, c, d. Figure 3 presents very sparse information, probably it is not needed.
Author Response
Response to Reviewer 2 Comments
Point 1. The MS starts with a good general description of greenhouse production, its problems and promises of biological and integrated pest management. It is very informative and nicely links the study in practice. However, I consider that strong arguments, such as distribution chains establishing “much more restrictions than those to community legislation itself”, would require references for support. It is noteworthy that in the end of MS, similar statements such as “current changes in legislation” and “need to increase sustainability of agriculture in terms of public health and environment” are again given with no references. I do believe that the described processes are ongoing, but the authors should provide explanation and their source of information.
Response 1: The reference… has been included.
Point 2 like the novel idea of looking for antagonists among pathogens of fungi. For some reason, I found no attempts of explaining the background of the approach in the introduction of the MS. General information of combative fungal interactions surely had been available in the literature. A problem with the MS is that it does not discuss the modes of interaction between plant and Trichoderma species nor fungal interactions. Figure 1 is very interesting as it shows Trichoderma aggressivum colonization on melon root. I am not aware of the possible mode of colonization, athough Trichoderma appears to promote plant growth. The authors could further clarify the interaction type, if it is known, or shortly review literature of such interactions. Also there is accumulating information of fungal interactions which could be used in explaining the antagonism type of reactions described here.
Response 2: There are no references to the use of fungal pathogens for the biological control of diseases at present. That is why it has been the subject of a patent. If bibliographic references have been included regarding the development of Trichoderma as biological control agents.
There are numerous references explaining the interactions between Trichoderma and plants or phytopathogens. Some of them have been included in this manuscript. But we do not consider that we hypothesize in these interactions because in this manuscript they have not been studied.
Point 3. The presentation of the results could be improved. The font size in figures 2,3 and 5 is too small to allow detailed examination in printed form. The different Trichoderma isolates could be illustrated with different bar color or splitting figures in a, b, c, d. Figure 3 presents very sparse information, probably it is not needed.
Response 3: The figures has been modified and statistical analysis has been included.

Reviewer 3 Report
After getting acquainted with the manuscript, I think that this work is very interesting and contains innovative elements. There is a strong need to look for biocontrol agents for pathogenic fungi. Well prepared manuscript.
Small errors:
line 139: Explain the abbreviation EC?
line 205 What does the abbreviation 'AT' mean?
Author Response
Point 1.
errors:
line 139: Explain the abbreviation EC?
line 205 What does the abbreviation 'AT' mean?
Response 1: Electrical conductivity has been included and AT is TA (T. aggressivum). The error has been eliminated.

Reviewer 4 Report
The present manuscript deals with an important current topic nowadays, which is the growing concern of citizens and consumers for food systems sustainability and food safety.
The title is misleading. It should be something like "Plant growth promotion and biocontrol ability of saline tolerant Trichoderma isolates under salinity stress". Maybe it should also be emphasized that Trichoderma isolates are effective against Pythium.
Why was Figure 1 included in the Materials and Methods section and not in the Results?
In Figures 2 and 3 the standard error bars are not visible. Furthermore, in these Figures significant differences have not been highlighted. In fact, these two figures are low resolution and the letters are too small to be read.
Throughout the manuscript, but particularly in the discussion, it is sometimes not obvious to which fungus the authors refer, either Pythium or Trichoderma. This is very important. Please go through the text and try to fix this.
The fact that Trichoderma strains may act as biocontrol agents against fungal phytopathogens is well established, however, the mechanisms are still not know. Please refer to a review, such as the one by Gajera et al. (2013). Therefore, the authors need to make very clear that these fungal pathogens are definitely not phytopathogens!
At the end of the discussion, the authors should leave their main findings to the end, i.e., they could perhaps change the order of the last two paragraphs.
Author Response
Response to Reviewer 4 Comments
Point 1. The title is misleading. It should be something like "Plant growth promotion and biocontrol ability of saline tolerant Trichoderma isolates under salinity stress". Maybe it should also be emphasized that Trichoderma isolates are effective against Pythium.
Response 1: The titule has been modified:
Plant growth promotion and biocontrol of Pythium ultimum by saline tolerant Trichoderma isolates under salinity stress.
Point 2. Why was Figure 1 included in the Materials and Methods section and not in the Results?
Response 2: The authors have decided to put Figure 1 on material and methods to explain better what fungi have been used in this study
Point 3. In Figures 2 and 3 the standard error bars are not visible. Furthermore, in these Figures significant differences have not been highlighted. In fact, these two figures are low resolution and the letters are too small to be read.
Response 3: The figures 2,3 4 have been modified and statistical analysis has been included. Figure 2 and 3 have been made jointly.
Point 4. Throughout the manuscript, but particularly in the discussion, it is sometimes not obvious to which fungus the authors refer, either Pythium or Trichoderma. This is very important. Please go through the text and try to fix this.
Response 3: This error has been corrected.
Point 4. The fact that Trichoderma strains may act as biocontrol agents against fungal phytopathogens is well established, however, the mechanisms are still not know. Please refer to a review, such as the one by Gajera et al. (2013). Therefore, the authors need to make very clear that these fungal pathogens are definitely not phytopathogens!
Response 4: The fact that the application of mycopathogenic agents as plant promoters does not generate lesions in melon, as well as in other plant species, explains that they can be considered non-phytopathogenic.
The reference Gajera et al. (2013) has been included.
Point 5. At the end of the discussion, the authors should leave their main findings to the end, i.e., they could perhaps change the order of the last two paragraphs.
Response 5: the authors have left for the end the recommendation, by way of final conclusion. The final paragraph would not make sense if it goes before.
